# Metabolic and Nutritional Characteristics of Long-Stay Critically Ill Patients

**DOI:** 10.3390/jcm8070985

**Published:** 2019-07-07

**Authors:** Marina V. Viana, Olivier Pantet, Geraldine Bagnoud, Arianne Martinez, Eva Favre, Mélanie Charrière, Doris Favre, Philippe Eckert, Mette M. Berger

**Affiliations:** 1Service of Adult Intensive Care Medicine and Burns, Lausanne University Hospital (CHUV), 1011 Lausanne, Switzerland; 2Service of Endocrinology, Diabetes and Metabolism, Lausanne University Hospital, 1011 Lausanne, Switzerland

**Keywords:** chronic critical illness, protein, Nutrition Risk Screening (NRS-2002), age, nutrition, vasopressors, shock, glucose, diabetes, underfeeding

## Abstract

Background: insufficient feeding is frequent in the intensive care unit (ICU), which results in poor outcomes. Little is known about the nutrition pattern of patients requiring prolonged ICU stays. The aims of our study are to describe the demographic, metabolic, and nutritional specificities of chronically critically ill (CCI) patients defined by an ICU stay >2 weeks, and to identify an early risk factor. Methods: analysis of consecutive patients prospectively admitted to the CCI program, with the following variables: demographic characteristics, Nutrition Risk Screening (NRS-2002) score, total daily energy from nutritional and non-nutritional sources, protein and glucose intakes, all arterial blood glucose values, length of ICU and hospital stay, and outcome (ICU and 90-day survival). Two phases were considered for the analysis: the first 10 days, and the next 20 days of the ICU stay. Statistics: parametric and non-parametric tests. Results: 150 patients, aged 60 ± 15 years were prospectively included. Median (Q1, Q3) length of ICU stay was 31 (26, 46) days. The mortality was 18% at ICU discharge and 35.3% at 90 days. Non-survivors were older (*p* = 0.024), tended to have a higher SAPSII score (*p* = 0.072), with a significantly higher NRS score (*p* = 0.033). Enteral nutrition predominated, while combined feeding was minimally used. All patients received energy and protein below the ICU’s protocol recommendation. The proportion of days with fasting was 10.8%, being significantly higher in non-survivors (2 versus 3 days; *p* = 0.038). Higher protein delivery was associated with an increase in prealbumin over time (*r*^2^ = 0.19, *p* = 0.027). Conclusions: High NRS scores may identify patients at highest risk of poor outcome when exposed to underfeeding. Further studies are required to evaluate a nutrition strategy for patients with high NRS, addressing combined parenteral nutrition and protein delivery.

## 1. Introduction

The intensive care unit (ICU) patient population has evolved over the last two decades with the appearance of an increasing number of patients requiring very long ICU stays, lasting up to several months after surviving the initial acute insult [1]. A long stay is usually defined as the requirement of more than one week of mechanical ventilation (MV) and of ICU therapy, but different definitions have been used [2]. Chronic critical illness (CCI) is the most frequent designation for these patients, which is characterized by lengthy hospital stays, intense suffering, high mortality rates, and substantial resource consumption [3]. Genetic influence has also been shown to be increasingly present, particularly in pediatric ICUs [4]. Preexisting chronic comorbidities are strong, independent predictors of this condition [5]. In the ICU, mechanical ventilation, sepsis, Glasgow score < 15, inadequate calorie intake, and higher body mass index (BMI) have been identified as independent predictors of this condition [6].

Recently, a study including 185 chronic, critically ill patients showed that the preadmission nutritional status reflected by the Nutrition Risk Screening (NRS-2002) (hereafter NRS) [7] might be a good predictor of the outcome: a score ≥ 5 points seemed to be a cutoff predictor of mortality [8]. Entering the acute disease with deficits such as weight loss, being older, or malnourished, seems to constitute a metabolic handicap that threatens the patient’s response capacity and survival. In 2013, the same group observed that inadequate nutrition (defined as the provision of less than 60% of needs) and organ failure (defined by the sequential organ failure assessment (SOFA) score) are mortality risk factors [6].

The optimal timing and amount of feeding has been much debated. Indeed, the intestine of the sickest patients is not always available to accommodate full feeding as shown by the French randomized controlled, multicenter, parallel group trial called NUTRIREA-2 including 2410 patients with septic shock [9]. Furthermore, many patients are at risk of the refeeding syndrome [10]. A recent Brazilian study showed in a cohort of 100 critically ill patients with a mean ICU stay of 19 days, that those 45 patients receiving a caloric intake ≥70% during the first 72 h of hospitalization did not present better outcomes in the short term (mechanical ventilation, length of ICU stay) or after 1 year (functional capacity, mortality) [11]. For both above reasons, i.e., risk of stressing the intestine and of relative overfeeding [12], the early full feeding strategy seems unable to promote a good outcome. In this context, the most recent European Society for Clinical Nutrition and Metabolism (ESPEN) guidelines propose increasing progressively (ramping up) the energy delivery whatever the feeding route over 3–4 days [13]. Furthermore, prescribed energy goals should be covered somewhere between day 4 and 7, and preferentially be determined by indirect calorimetry.

To better coordinate the treatment of the long stay patients admitted to our ICU, and to identify procedures that could be improved, the service initiated a dedicated program enrolling patients requiring more than 2 weeks of ICU treatment. The present study aims to describe the metabolic and nutritional specificities of these patients, to analyze those associated with poor outcome, and to identify a factor that might enable early detection of CCI risk.

## 2. Methods

With the approval of the Commission Cantonale d’Ethique sur la Recherche humaine (CER 2018-02018), consecutive patients admitted to the 35 bed multidisciplinary ICU of the Lausanne University Hospital were analyzed. The patients had been prospectively enrolled in the long-stay program of the ICU, a plan of action called PLS (Patients Long Séjour) created in January 2017 to improve the care of patients requiring more than 2 weeks of ICU therapy (clinical trial identifier: NCT03938961). The PLS program consisted of weekly interdisciplinary meetings addressing ventilation, nutrition, cognitive and functional issues, symptom management, psychological support, and complication prevention. The meetings ended with detailed evaluation and a plan of action.

### 2.1. Patients

The inclusion criteria were age >18 years, and inclusion in the PLS program with an ICU stay >2 weeks. The exclusion criteria were admission for major burns >20% body surface, traumatic brain injury, and patients who refused participation.

### 2.2. Study Variables

Data were extracted from the computerized information system (Metavision® iMDsoft, Tel Aviv, Israel, version 5.46.44) and included age, admission (dry weight) and discharge body weight, body mass index (BMI), NRS-2002 score [7], severity of disease (SAPSII), Sequential Organ Failure Assessment (SOFA) score on days 1, 2, 5, 10, 30, and discharge [14]), requirement of continuous renal replacement therapy (CRRT), admission for sepsis, presence of diabetes, daily intakes (energy, protein, glucose: raw data and per kg), all arterial blood glucose values, 24 h insulin dose, arterial blood lactate (maximal value), daily feeding route, pressure ulcers occurring during the ICU stay, discharge Medical Research Council force score (MRC), length of mechanical ventilation, ICU and hospital length of stay, ICU and 90-days mortality. Laboratory: blood prealbumin (determined weekly), C-reactive protein (CRP), and procalcitonin (mean of the ICU stay).

### 2.3. Nutrition Protocol

The ICU’s protocol called NUTSIA (NUTrition aux Soins Intensifs Adultes) is based on the ESPEN guidelines [13]. Enteral nutrition is recommended as the first option when nutritional therapy is indicated (i.e., for patients for whom oral intake is not possible), and should be initiated within the first 3 days after hemodynamic stabilization. Parenteral nutrition (PN) may be used as combined feeding, or total PN from day 4 and earlier in selected malnutrition situations. Energy goals are 20 kcal/kg/day during the first week, to be increased or adapted thereafter—indirect calorimetry is recommended from day 7. The initial energy goal should be reached by day 4–5. Energy intakes include the energy from feeding products and oral intakes, and the energy resulting from glucose (drug dilution) and fat (sedative propofol), which are administered for non-nutritional purposes.

Feeding products were: Peptamen Intense (Nestlé, Switzerland), Promote Fiber plus (Abbott, Switzerland), Isosource Energy Fiber (Nestlé, Switzerland), Nutriflex Omega special (BBraun, Switzerland), and Pharmacy compounded PN in individual cases.

Energy balances were calculated as the daily difference between the total substrate intakes (nutritional and non-nutritional sources) and the prescribed value. The mean daily and the cumulated values were calculated for the first 10 days, and a cutoff of cumulated deficit −70 kcal/kg was considered critical [15,16]. For proteins, the daily delivery was compared to the recommended target of 1.2 g/kg/day. The cumulated deficit was calculated for the first 10 days and a cutoff of −300 g was considered critical.

Glucose control is handled by nurses, with a blood glucose (BG) goal of 6–8 mmol/L (6–10 mmol/L in diabetic patients) maintained using continuous insulin infusion. Blood glucose is presented as the daily mean value, and is measured on a point of care blood gas analyzer. Its variability is indicated as the daily standard error of the daily BG values. Daily total insulin was recorded.

The 2 dieticians (one full time position present on working days) attended to patients requiring >3 ICU days. These individuals are in charge of checking route of feeding, energy needs (indirect calorimetry), proteins delivery, and adapting the feeding solutions [17]. They attend the weekly PLS meetings.

### 2.4. Other Variables

Muscle strength upon discharge was assessed using the MRC score [18], where the patient’s effort is graded on a *scale* of 0–5. Six muscle groups were bilaterally measured (abduction of the arm, flexion of the forearm, extension of the wrist, flexion of the hip, extension of the knee, and dorsal flexion of the foot). All muscle groups were scored between 0 and 5 (0, no visible/palpable contraction; 1, visible/palpable contraction without movement of the limb; 2, movements of the limb but not against gravity; 3, movements against gravity (almost full passive range of motion) but not against resistance; 4, movement against gravity and resistance, arbitrarily judged to be sub-maximal for gender and age; 5, normal). Therefore, the maximal score is 60 points.

### 2.5. Statistical Analysis

Data of the entire stay were extracted, but analysis was limited to the first 30 days. The data are presented as means ± standard deviation (SD), or median (interquartile Q1, Q3) depending on normal distribution, which was assessed through histograms and calculation of skewness and kurtosis. For non-parametric variables, the Kruskal-Wallis test was used. Two phases were considered for the analysis of the nutritional related variables: the first days (D1–D10), and the next 20 days of the ICU stay (D11–D30). The continuous variables in survivors versus non-survivors were compared using the one-way and two-way ANOVA, while the categorical variables were compared using Chi^2^ tests. For non-parametric variables, Kruskal-Wallis rank sum test was used. Kaplan–Meier analysis and Cox regression were used to compare mortality. For the Kaplan–Meier analysis, the log rank test was used to compare the curves of each NRS group (3–4 and 5–6–7 points). The generalized linear model (GLM) was used to analyze response variables that were not normally distributed such as substrate administration or blood glucose over the 30 days. Statistical programs were JMP version 14.2 for Windows, (SAS Institute GMH, Böblingen, Germany), and R Version 3.5.3, 2019 (R Foundation for Statistical Computing, Vienna, Austria). Significance level was set at *p* < 0.05.

## 3. Results

### 3.1. Patients

The current study included prospectively 150 patients admitted between 1 February 2017 and 31 December 2018. During the same period, 589 patients were admitted who required ICU for >7 days, but the majority were discharged before entering the PLS program. Table 1 summarizes patient demographics and clinical outcomes, with the details of survivors/non-survivors at ICU discharge. Most patients (60.7%) were admitted for medical causes (17/91 with a surgical background), followed by emergency surgery, and elective surgery (52% were admitted for sepsis, and 48% needed continuous renal replacement during their stay).

Gender was equally distributed, but the non-survivors were older (*p* = 0.024) and tended to have a higher SAPSII score (*p* = 0.072). The admission SOFA score did not differ significantly but its evolution was different over time, decreasing significantly in survivors (*p* < 0.0001) (Figure A1). The cardiac component of the score remained elevated for a longer period of time (i.e., at 3–4 points) in the non-survivors. A significantly higher NRS score (*p* = 0.033) was observed in non-survivors. The Kaplan–Meier analysis showed that the scores NRS 5–6–7 are associated with a higher 90-days mortality of 43.8% versus 25.7% with NRS scores 3 and 4, respectively (Figure 1). Admission weight (mean 78.3 kg) and BMI were similar, but the weight change over time differed significantly (*p* < 0.0001) depending on the outcome. By the end of the stay, survivors had lost weight, while non-survivors had gained weight due to a persistent positive fluid balance. While 28 patients (18%) died in the ICU, further 26 patients died within 90 days, resulting in a total of 56 (35.3%) deaths—age and NRS differences became even more pronounced by day 90 (age *p* < 0.001; NRS *p* = 0.005; Table A1). A Cox regression model with 90-days mortality as an outcome showed an increased risk of death in patients with NRS ≥ 5 (HR 2.2 (1.18–4.2), *p* = 0.013), but not for the SAPS2 score (HR 1.0 (0.99–1.0), *p* = 0.235). Diabetes mellitus was present in 17.3% of patients, but was not associated with any mortality difference.

### 3.2. Nutrition

The proportion of days with intentional absence of feeding (fasting) was 10.8% of total days (Table 2 and Figure 2), being significantly higher in non-survivors (*p* = 0.038), with fasting contributing to the high day-to-day variability of intakes (Figure 3)—except for the number of fasting days (3 versus 2, *p* = 0.043). Table 2 shows that there was no difference between survivors and non-survivors. Fasting days were mostly observed during the first 5 days, but did occur throughout the 30 days. The feeding route was predominantly enteral, representing 55.7% of total days; combined enteral (EN) and parenteral (PN) feeding represented 8.1% and 14.1% of total days, respectively, and variable route combinations was used in 1% of total days.

#### 3.2.1. Energy

The mean prescribed energy during the first 10 days was 21.7 kcal/kg/day, and 23.4 kcal/kg/day thereafter (1750 kcal/day). The individual patients were characterized by a high variability of feeding, and total protein and glucose intakes (Figure 3). The progression of feeding occurred over the first 10 days (and not over 5 days as per protocol), i.e., before inclusion in the dedicated program (no difference between the two categories). Figure 4 shows the mean values for protein and energy, which were below the service’s recommendations and below the prescribed values. Both groups were underfed, resulting in a median cumulated negative balance of −5266 kcal by day 10 (−70 kcal/kg). Nutrition delivery improved over time, but stagnated below the prescribed value.

Indirect calorimetry was available in 95 patients. The prescription was −236 kcal (range: −1391 to +483 kcal) below the measured energy expenditure. When this deficit is added to the daily energy deficit (difference between prescribed and delivered), the median deficit becomes −741 kcal/day (range: −2971 to −123 kcal per day) for the first 10 days.

#### 3.2.2. Proteins

The delivery was lowest during the first 10 days (Figure 4), with a median of 64.3 g/day (0.70 g/kg/day), which is below the 1.2 g/kg/day service NUTSIA recommendation. By day 10, 80 patients (53.3%) exceeded −300 g of cumulated deficit. Protein intake thereafter increased to 1 g/kg/day remaining below recommendation, with a tendency to be lower in non-survivors.

Prealbumin, the lowest values were observed in patients with high NRS scores (*p* = 0.067). Prealbumin increased over time in the majority of patients in association with higher protein delivery (*r*^2^ = 0.19; *p* = 0.027), and the increase (difference between first and last value of the ICU stay) was lower in non-survivors (*p* < 0.001).

#### 3.2.3. Blood glucose

Altogether, 30,769 arterial blood glucose values were available for analysis (4–10 blood samples per day). Blood glucose and related variables differed between survivors and non-survivors (Figure 5), and over time. While glucose intakes were identical and low, non-survivors had higher BG values during the first 10 days, but not thereafter. Variability was higher, but not significant. Insulin needs were significantly higher and became again higher after day 20, while survivors’ insulin needs declined, reaching below 30 u/day. Blood maximal lactate was also significantly higher during the first days in non-survivors.

The BG pattern differed in diabetic patients, being higher throughout the stay as per protocol, with higher variability and higher 24 h insulin requirements (Figure A2). This was particularly marked in surviving diabetic patients (Figure A3). Diabetic non-survivors were characterized by lower BG values and higher insulin needs during the first 5 days. Arterial lactate was elevated during the first 5 days in the majority of patients, being significantly higher in diabetic non-survivors.

### 3.3. MRC Discharge Score

MRC discharge score was available in 91 patients. The median value on discharge was low with 34 points. In those patients who did not suffer significant persistent renal failure on discharge (which results in elevated creatinine values), the median serum creatinine was 52 µmol/L (a low value in adult patients, and a surrogate for muscle mass).

## 4. Discussion

The main finding in our study is that CCI patients are not equal upon admission. Many patients begin their ICU journey without metabolic reserves reflected by a high NRS score. However, the impact is not immediate. During the first 10 days, the patients were exposed to very low and highly variable energy and protein intakes, and many were exposed to prolonged and repeated fasting. The acute underfeeding generated important energy and protein deficits, resulting in an additional malnutrition diagnosis in these critically ill patients. Combined EN and PN feeding was minimally used, despite being included in the service’s protocol based on randomized trials [19,20]. In patients at risk of malnutrition upon admission, the underfeeding further erodes a previously altered metabolic status, resulting in a poor outcome. These results confirm recent observational data [16], where patients at low risk are likely to have a reasonable outcome despite several days of inadequate nutrition, while those at higher risk do not. The 2019 ESPEN recommendations [13] emphasize that no patient should be left longer than 72 h without initiating a balanced feeding. The majority of nutrition-related studies focus on the first days after admission, and have often included moderately sick or young patients that did not really benefit from artificial nutrition. By extracting the data of the entire stay, we observed that there was an early phase, and a later phase with a turning point around day 10. Thereafter, the patients were better fed, but the majority did not reach their prescribed goals.

The patient cohort was critically ill, as reflected by an elevated admission SAPSII score and an 18% ICU mortality, which was significantly higher than the general ICU patient’s mortality (14.4%). The non-survivors were older, which belongs to the risk factors of CCI [1]. At the 90 days time point, a further 26 patients had died, increasing the total mortality to 35.3%. The initial SOFA score did not differ, but the score’s evolution differed in survivors and non-survivors, remaining higher in the latter. The inclusion of the evolution of organ failure over time assessed by the SOFA score is an important strength of this study. Our data do not confirm those of a Brazilian cohort of 453 CCI patients, in which the initial SOFA score was a predictor of mortality [6]. One of the aims of the study was to find a variable facilitating the early detection of patients at critical illness risk. The total SOFA score did not predict the outcome, while the NRS score was able to detect them, confirming other Brazilian results [8]. A NRS score ≥ 5 should raise an alert, especially when associated with hemodynamic instability and high lactate values.

Timing and route of feeding in the critically ill has long been debated, and particularly the timing of PN initiation. The pragmatic CALORIES trial in 2400 mechanically ventilated patients randomized to EN or PN from day 1 showed no difference between the two routes [21]. The NUTRIREA-2 study included 2410 patients with septic shock, and showed that full early EN from day 1 was associated with more intestinal complications (*p* < 0.001) compared to the same dose of PN [9], with no benefit of EN on infectious complications. Providing early full feeding does not seem to prevent patients from becoming CCI, as shown by a Brazilian study including 100 critically ill patients. The study further shows that providing a caloric intake ≥70% in the first 72 h of admission did not improve short term or 1-year outcomes [11]. Among the possible explanations of these disappointing results, the early endogenous glucose production [20,22] in response to acute illness should be considered. This production covers approximately two thirds of energy expenditure during the first days of disease. In addition, enteral full feeding is not always tolerated, as shown by NUTRIREA-2 [9], and the risk of refeeding syndrome is a reality [10]. The dose of both feeding and vasopressors might be the explanation as shown by the secondary analysis of a large septic cohort which showed that EN providing close to recommended intakes of energy and proteins over the stay was associated with a better outcome [23]. A propensity-matched analysis, including 52,563 ventilated adult patients stratified by dose of noradrenalin [24], suggests that in patients on low- or medium-dose noradrenalin, early EN seems associated with a reduction in mortality but not in the patients requiring high-dose noradrenalin. The recommendation to provide early, but progressive feeding starting ideally during the first 48 h, while discouraging early full feeding whatever the route, formulated in the latest ESICM [25] and ESPEN [13] guidelines was built on such data.

The ICU’s internal protocol recommends progressing to the goal by day 4–5 based on ESPEN recommendations, which was not achieved without any difference between survivors and non-survivors. The latter were characterized by a higher percentage of fasting days during the first days after admission (associated with hemodynamic instability), but also occurring at random thereafter. The reasons for the feeding interruptions are multiple in an ICU, and have not changed much over the last decades [26]. The elevated mean cardiovascular SOFA scores reflect a persistent hemodynamic instability, which is probably one of the reasons for withholding feeding, and for difficult EN progression. These poor results may need a change in our ICU feeding procedure. Volume based feeding needs consideration as recent studies show that it is safe, and effectively improves energy and protein delivery compared to traditional rate-based feeding [27].

An American study analyzed the discharge destinations of critically ill surgical patients according to the magnitude of their energy and protein deficit. The authors showed that nutrition was a major outcome determinant [16]. Yeh et al. used a cumulated deficit cutoff of −6000 kcal and of −300 g protein. The patients who remained below those cutoffs were three times more likely to be discharged home. The authors also observed a longer ICU stay and higher mortality in those patients exceeding these values. In the present cohort, the patients entered the PLS program only after 2 weeks. Large energy and protein deficits had built up, but did not differ between survivors and non-survivors, both being similarly underfed by day 10: 80 patients (53.3%) exceeded the −300 g protein deficit and 50 patients (33%) exceeded the −6000 kcal cutoff. Knowing that the indirect calorimetry values for energy expenditure were higher than prescription in the majority of patients, the real deficit is even greater. Malnutrition causes loss of lean body mass, which is a determinant of the outcome [28,29], with the early protein deficit being an important contributor to the loss. The positive response of prealbumin to higher protein delivery shows that increasing proteins delivery is a treatment option. Critical illness is characterized by a high degree of stress with an accelerated protein degradation that results in malnutrition, systemic inflammation, and organ dysfunction [30]. Supporting this hypothesis, Briassoulis et al. showed that in critically ill children, only 22.7% of patients without protein deficiencies versus 37% of those at risk or already deficient, developed multiple-organ system failure [31]. Further transferrin and prealbumin levels increased already after 5 days of early EN, and the patients with positive nitrogen balance had higher prealbumin levels [32]. In critically ill patients, this muscle loss occurs very rapidly as shown in the landmark study by Puthucheary et al. [33]. In patients with two or more organ failures, the mean loss of muscle measured by ultrasound of the thigh was 22% in 10 days. The present patients were in multiple organ failure as reflected by their high SOFA score. However, there is currently no standardized procedure to assess sarcopenia in long-stay catabolic patients [34], and only surrogates are available: low (33 points) muscle strength at discharge (MRC score) and the discharge creatinine values below those recorded at admission reflect the loss of muscle mass. These results highlight the importance of an early multidisciplinary awareness for metabolic and nutritional issues, i.e., waiting 2 weeks to address them is too long. The timing, dose, and route of feeding must be addressed more stringently and earlier, and the recommendations of the dieticians applied more diligently in the high risk patients already by day 3–4.

Among the factors associated with ICU weakness, hyperglycemia has been considered important, as insulin therapy might be an attenuating factor of muscle loss through its anabolic properties [35]. Hyperglycemia is associated with increased mortality in critically ill patients [36], an issue that appears to improve by tight glycemic control [37]. The debate regarding optimal glycemic control continues as diabetic patients are frequent and exhibit a different response to tight glucose control [38]. However, there is little data in patients with prolonged ICU stay. One study reported that tighter glycemic control was associated with improved outcomes in CCI patients with stress hyperglycemia, but not in CCI patients with diabetes [39]. In the present study, the initial BG values were elevated, independently of low glucose intake. BG normalized around day 10, and remained stable thereafter, being higher in the non-survivors despite similar glucose intakes. The observed changes in BG and insulin needs over time may be related to the decrease in lean body mass. The future non-survivors also required more insulin during the 2^nd^ phase, possibly reflecting a metabolic derangement. Diabetic patients exhibited a different pattern compared to non-diabetics corresponding to the application of the internal protocol.

Limitations of the study: our current study is observational, including only 150 patients, but the patients were enrolled prospectively to the PLS program and the study was aimed at identifying early risk criteria. The strength of the data comes from the quality of data extraction from the computer system, resulting in longitudinal continuous daily data. Regarding fasting, the study was not designed to record the reasons for the feeding interruptions, so exact causes are missing but are unlikely to differ from other observations [26]. Furthermore, our study lacks a measure for the loss of lean body mass. Except for ultrasound, which was not available in our clinical setting, there is currently no standardized procedure to assess sarcopenia in CCI patients [34]. Whole body multi-frequency bioimpedance, phase angle, and ultrasound assessment of the cross-sectional area of the thigh might become clinically useful tools, as they are non-invasive and little costly. We therefore consider integrating them into the nutrition protocol.

In randomized trials, our group showed that, in case of insufficient enteral feeding during the first 3 days (i.e., <60% of 25 kcal/kg/day which is <15 kcal/kg/day), supplemental PN (SPN) guided by indirect calorimetry to cover the measured value from day 4, reduced nosocomial infections [19] and related costs [40]. Furthermore, our group recently showed that this reduction was explained by modifications of the immune response and attenuation of the inflammatory response [20], with a trend to less loss of muscle mass (*p* = 0.07). This combined strategy should probably be applied more frequently, especially in patients with NRS ≥ 5 who obviously are not covering their needs, and who have persistent high cardiac SOFA scores. The two above Swiss SPN studies also show that a two-phase approach may be the most respectful of physiological mechanisms. Critically ill patients are generally unstable during the first days, and their endogenous energy production is elevated [22], providing 200–300 g glucose per day. This endogenous glucose production has an elevated protein cost, and 120 g/day were shown to be catabolized for neoglucogenesis. Full feeding during this period is therefore associated with a risk of overfeeding [12]. The present study shows that the very simple NRS score is probably able to identify patients who will not tolerate a prolonged underfeeding. This two-step, more physiological approach and the absence of difference between the enteral and parenteral feeding was recently summarized [41]. It is also important to integrate that “nutrition is more than the sum of its parts” as described by Briassoulis et al. [42], who emphasized that focusing only and separately on proteins or energy is not sufficient. An integrated, individualized approach, including all substrates and micronutrients is needed [43].

## 5. Conclusions

The present study shows that high NRS scores upon admission may identify patients at higher risk of a poor outcome, and those who require individualized nutrition therapy [43]. Exposing these high-risk patients to underfeeding further deteriorates their response capacity, including immune defenses [40], possibly favoring the development of chronic critical illness. The NRS score is not a malnutrition score, but identifies those patients who are at risk of high mortality at an early stage, this risk increasing further with underfeeding. Having no reserves, these patients will cope worse, and continue eroding their lean body mass, contributing to the poor prognosis [28,29]. Our study also shows that the most unstable patients, who become chronic, are exposed to discontinuous feeding, partly explained by persistent shock and high blood lactate values which question the clinicians about intestinal integrity, and the feasibility of EN [25]. In our previous studies [15], and in others [16], the cutoff of −70 kcal/kg of cumulated energy deficit was associated with increasing complications. Preventing such a deficit to build up requires identifying the patients at risk by their elevated NRS and hemodynamic instability. This would enable the introduction of combined feeding by day 4–5, while checking the real needs by indirect calorimetry. This strategy should be tested prospectively.

## Figures and Tables

**Figure 1 jcm-08-00985-f001:**
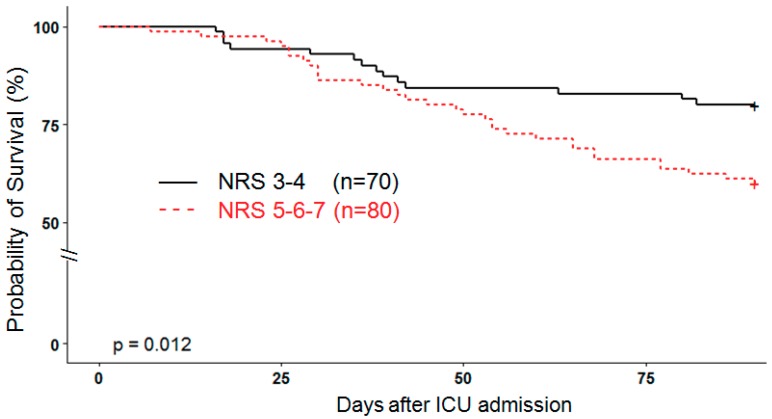
Kaplan–Meier analysis comparing elevated and low NRS scores. NRS: Nutrition Risk Screening and ICU: intensive care unit.

**Figure 2 jcm-08-00985-f002:**
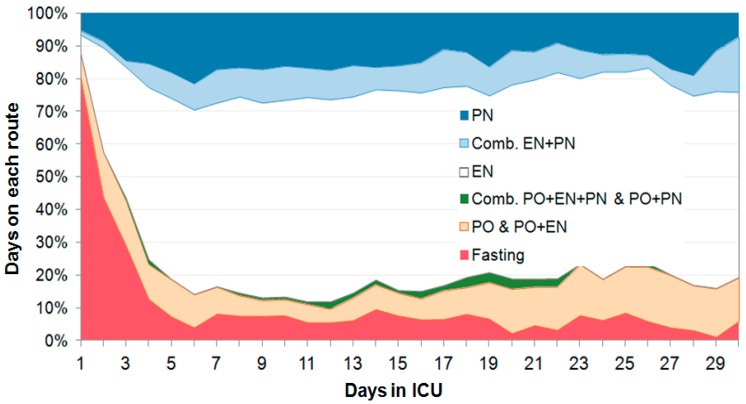
Evolution of the route of feeding over time presented as percentage of all patients over the first 30 days—there is a variable time of fasting during the first week. Enteral feeding was predominant, with a stable proportion of combined enteral–parenteral feeding (Comb EN + PN), or total parenteral nutrition (PN), and a variable proportion of the combinations oral–enteral, or oral–parenteral. Abbreviations: EN = enteral nutrition, PN = parenteral nutrition, Comb = combined, PO = oral.

**Figure 3 jcm-08-00985-f003:**
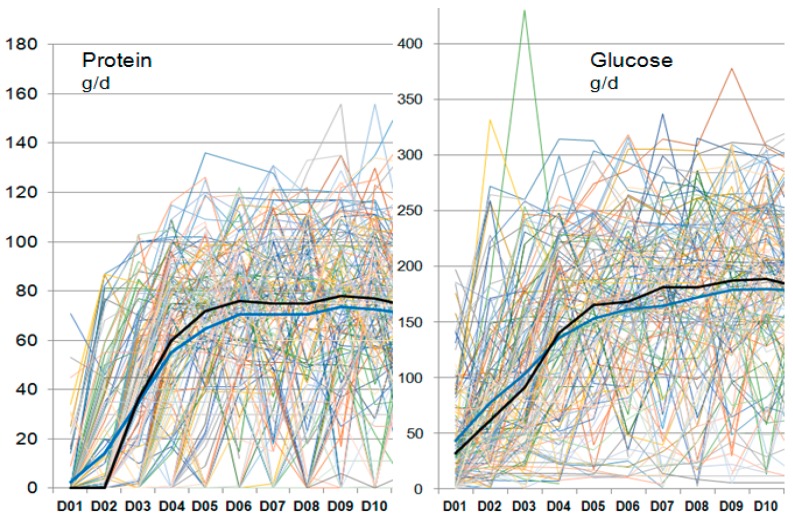
Individual delivery of protein and glucose by day during the first 10 days. Each line represents individual patient values. The erratic aspect aims at showing a phenomenon which is the extreme day to day variability and multiple interruptions that characterize the nutrition in the early phase. The thick dark lines show median (blue) and mean (black) values.

**Figure 4 jcm-08-00985-f004:**
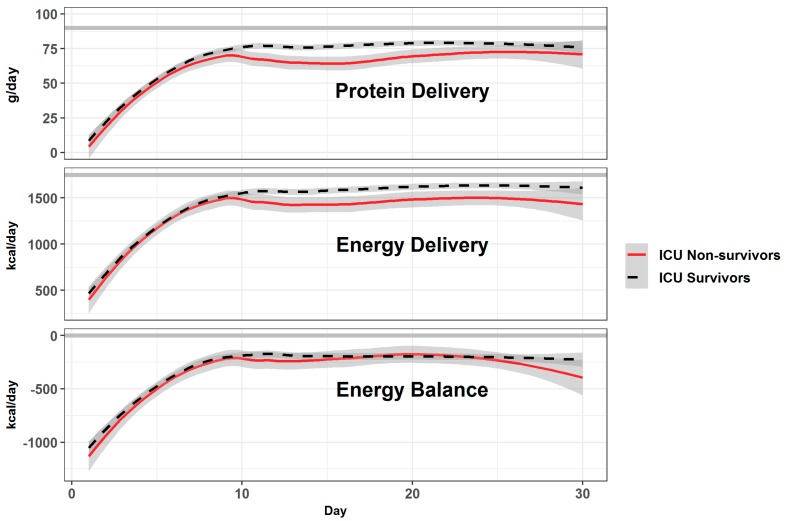
Mean protein and energy delivery with the resulting energy balance over the first 30 days according to ICU vital status (mean ± SD). The thick gray lines show protein target (1.2 g/kg/day), energy goal (prescribed value), and neutral energy balance. The differences in protein and energy delivery between survivors and non-survivors were significant after day 10 (*p* < 0.001). Energy balances were similarly negative.

**Figure 5 jcm-08-00985-f005:**
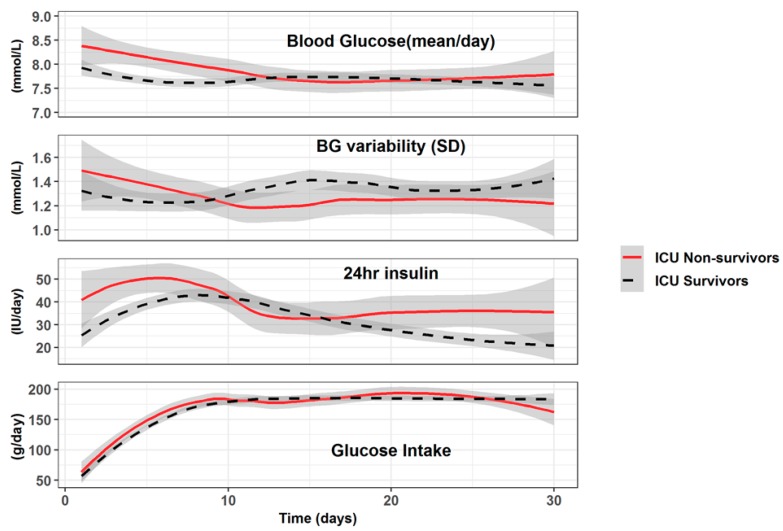
Evolution of mean blood glucose (mmol/L), Blood Glucose (BG) variability (standard error = SD of the individual daily values), 24 h insulin (total dose/24 h), and glucose intake (total in g/day). The figure shows that the 24 h Insulin delivery is not related to the glucose intake, which was similar in both groups (data are represented as mean ± SD).

**Table 1 jcm-08-00985-t001:** Demographics, severity of illness, laboratory and outcome variables according to ICU vital status.

	Overall	Non-Survivors	Survivors	*p*
N (%)	150	27 (18%)	123 (82%)	
Age (mean (SD))	60.2 (14.6)	66.0 (10.9)	59.0 (15.0)	0.024
Sex (Males, %)	116 (77.3)	24 (88.9)	92 (74.8)	0.184
Body weight admission (kg, mean (SD)) discharge	78.3 (18.4) 75.7 (17.8)	77.4 (18.8) 81.5 (18.5)	78.4 (18.4) 74.3 (17.5)	0.785 0.054
BMI (mean (SD))	26.47 (6.09)	26.15 (6.42)	26.54 (6.05)	0.761
SAPS2 (mean (SD))	52.9 (18.3)	58.7 (15.8)	51.7 (18.6)	0.072
SOFA on Day1 (median (Q1, Q3))	8 (5, 11)	11 (9, 14)	9 (6, 12)	0.134
NRS (median (Q1, Q3))	5 (3, 6)	5 (4, 6)	4 (3, 6)	0.033
Medical/Emergency surgery/elective surgery	91/40/19	20/6/1	71/34/18	0.141
Diabetes (%)	26 (17.3)	7 (25.9)	19 (15.4)	0.307
Renal failure requiring CRRT (*n*, % of *n*)	72 (48.0%)	22 (81.5%)	50 (40.5%)	<0.001
Sepsis on admission (*n*, % of *n*)	78 (52.0)	15 (55.6)	63 (51.2)	0.845
CRP (median (Q1, Q3)) mean of stay	125 (81,170)	153 (97, 183)	116(77, 166)	0.096
Procalcitonin (median (Q1, Q3)) mean of stay	1.4 (0.4, 4.6)	3.2 (1.6, 7.7)	1.0 (0.3, 3.7)	0.008
Glucose (daily means (median (Q1, Q3))	7.7 (7.2, 8.4)	7.8 (7.4, 8.2)	7.6 (7.1, 8.4)	0.408
Lactate (daily max (median (Q1, Q3))	1.7 (1.4, 2.1)	2.0 (1.7, 2.7)	1.6 (1.4, 2.1)	0.004
Pressure ulcers (n patients with)	72 (48%)	12	60	0.835
MRC on discharge (median (Q1, Q3) (*n* = 91))	34 (24,42)	31 (25, 43)	35 (25,42)	0.825
Length of Mech.Ventilation (days) (median (Q1, Q3))	16.1 (10.0, 21.6)	17.8 (13.2, 26.8)	15.7 (9.8, 20.3)	0.152
Total length of ICU stay (median (Q1, Q3))	31 (23, 46)	29 (18, 44)	31 (24, 46)	0.268
Hospital length of stay (median (Q1, Q3))	57 (39, 82)	30 (24, 47)	63 (44, 91)	<0.001

Abbreviations: SOFA = Sequential Organ Failure Assessment, SAPS = Simplified Acute Physiology Score, BMI = Body Mass Index, NRS = Nutrition risk screening, CRP = C-reactive protein, Q1 and Q3 = q quartiles 25, 75.

**Table 2 jcm-08-00985-t002:** Nutrition characteristics according to ICU outcome (D = day).

	Overall	Non-Survivors	Survivors	*p*
N (%)	150	27 (18%)	123 (82%)	
Days of fasting: N per patient (median (Q1, Q3))	2.0 (1.0, 3.0)	3.0 (1.0, 4.0)	2.0 (1.0, 3.0)	0.043
Percentage of days (median (Q1, Q3))	5.4 (2.4, 10.0)	7.8 (4.3, 10.9)	4.9 (2.0, 9.3)	0.031
Prealbumin (delta of stay) (median (Q1, Q3)) g/L	0.07 (0.04, 0.12)	0.06 (0.02, 0.10)	0.07 (0.04, 0.13)	<0.001
Energy delivery D1–10 (median (Q1, Q3)) kcal/day (median (Q1, Q3)) kcal/kg/day	1161 (957, 1370) 15.8 (11.8, 18.9)	1121 (936, 1385) 16.4 (12.1, 17.9)	1161 (983, 1368) 15.7 (11.8, 19.1)	0.719 0.838
Energy delivery D11–30 (median (Q1, Q3)) kcal/day (median (Q1, Q3)) Kcal/kg/day	1559 (1368, 1762) 20.8 (17.9, 23.9)	1504 (1284, 1645) 20.2 (15.5, 21.9)	1581 (1387, 1772) 20.9 (18.1, 24.1)	0.104 0.151
Cumulated Energy balance D1–10 (median (Q1, Q3)) kcal/day (median (Q1, Q3)) kcal/kg/day	−5266 (−8365, −2697) −70 (−102, −37)	−5365 (−9208, −2852) −74 (−125, −38)	−5234 (−8043, −2651) −69 (−101, −36)	0.519 0.345
Cumulated Energy balance D1–30 (median (Q1, Q3)) kcal /day (median (Q1, Q3)) kcal /kg/day	−7700 (−11,607, −4702) −96.8 (−148.2, −59.7)	−7710 (−12,197, −5097) −92.4 (−151.1, −67.1)	−7677 (−11,350, −4554) −97.5 (−146.9, −58.4)	0.532 0.801
Protein delivery D1–10 (median (Q1, Q3)) g/day (median (Q1, Q3)) g/kg/day	53.7 (40.7, 64.3) 0.69 (0.52, 0.86)	54.0 (40.2, 61.9) 0.73 (0.57, 0.86)	53.4 (41.7, 65.7) 0.68 (0.50, 0.87)	0.673 0.768
Protein delivery D11–30 (median (Q1, Q3)) g/day (median (Q1, Q3)) g/kg/day	75.4 (62.0, 90.3) 1.0 (0.6, 1.4)	70.0 (49.4, 83.7) 0.95 (0.68, 1.1)	76.4 (63.3, 90.4) 1.0 (0.8, 1.2)	0.051 0.104
Cumulate protein balance D1–10(median (Q1, Q3)) g/day (median (Q1, Q3)) g/kg/day	−374 (−595, −223) −4.86 (−6.82, −3.36)	−352 (−538, −244) −4.63 (−6.18, −3.49)	−379 (−608, 216) −4.98 (−7.07, −3.34)	0.803 0.938
Cumulate protein balance D1–30 g/day (median (Q1, Q3)) g/kg/day	−603 (−1070, −304) −7.91 (−12.47, −4.13)	−531 (−1006, −355) −7.89 (−13.62, 4.74)	−611 (−1069, −299) −8.43 (−12.34, −4.11)	0.912 0.816

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
