# Peer review of "Metabolic and Nutritional Characteristics of Long-Stay Critically Ill Patients"

_jcm, 2019, doi:10.3390/jcm8070985_

Round 1
Reviewer 1 Report
The study “Metabolic and Nutritional characteristics of long-stay critically ill patients” by Marina V Viana et al. aimed to describe the demographic, metabolic and nutritional specificities of patients prospectively enrolled to the long-stay program. The study concluded that high NRS scores might identify the patients at the highest risk of poor outcome when exposed to underfeeding.
The study is wall organized and written, having suffered from limitations, which authors adequately report at the end of their discussion. I have some important suggestions or comments to the authors, which I believe, if answered, will improve their article and enhance their messages to the readers.
Comments
1.37. It is characterized by lengthy hospital stays, intense suffering, high mortality rates and substantial resource consumption [2].
Chronic critical illness (CCI) has also been characterized by an increasing trend of recognized genetic influence. Please add this info and the reference (J Crit Care. 2005 Jun;20(2):139-46. Current trends of clinical and genetic characteristics influencing the resource use and the nurse-patient balance in an intensive care setting.)
1.38. Mechanical ventilation, sepsis, Glasgow score <15, inadequate calorie intake, and higher body mass index (BMI) have been identified as independent predictors of this condition [3].
It is important to refer to a study showing that chronic comorbid illnesses have been also identified as strong independent predictors of this condition (J Intensive Care Med. 2018 Oct 1:885066618783175. Chronic Comorbid Illnesses Predict the Clinical Course of 866 Patients Requiring Prolonged Mechanical Ventilation in a Long-Term, Acute-Care Hospital.)
5.142. Table 2. E
Energy presents as a balance (Cumulated Energy balance D10 (vs prescription)) on D10 while protein as intake only on D1-10 confusing the reader (Protein delivery D1-D10 (median [IQR]) g/day). It could be better to follow the same rules for energy and protein, preferably balance for both. In Table 2 we do not know if more protein has been prescribed than delivered
5.144. Figure 2. Evolution of the route of feeding over time: there is a variable time of fasting during the first week, enteral feeding is predominant, with a stable proportion of combined enteral-parenteral or total parenteral nutrition. Except for the number of fasting days (3 versus 2, p=0.048) there is no difference between survivors and non-survivors. Fasting days predominate early but occur through the 30 days.
According to the legend fasting days are presented and differences between survivors and non-survivors. I can see neither of these two. Also, where is PN (parenteral nutrition)? IV usually refers to a simple intravenous solution (Gl5%, NaCl 0.9% with electrolytes, etc). Abbreviations should be defined at the end of the legends.
6.149. Figure 3. Individual prescription of energy, and administration of proteins and glucose by day
Where is energy? Also, many patients were receiving each day 0 protein and 0 glucose (g/day). Are you sure? Something is wrong with this figure: could these recordings have actually not been by day but at a moment “snapshots” (i.e. 8 a.m.) of a day? You should re-check your data and even delete this figure.
7.166. Figure 4. Energy balance, energy and protein intake over the first 30 days according to ICU vital status (mean ± SD). The thick gray lines show energy goal (prescribed value), and, protein target (1.2 g/kg/day).
Again, one is balance the other intake! One goal is prescribed the other is not prescribed but fixed value. Energy intake is not presented as the legend indicates.
8.188. Figure 5. Evolution of blood glucose, BG variability (standard error = SD), glucose intake and 24hr insulin (mean ± SD). Insulin delivery is independent of glucose intake.
Units should be shown
9.242. The non survivors were characterized by a higher percentage of fasting days, principally during the first days after admission, but also occurring during at random during the stay.
It is of worth to refer here to recent studies showing that volume-based feeding safely and effectively improves energy and protein delivery to critically ill patients compared to traditional rate-based feeding and might improve patient outcomes. (Crit Care. 2019 Apr 2;23(1):105. doi: 10.1186/s13054-019-2388-7. Safety and efficacy of volume-based feeding in critically ill, mechanically ventilated adults using the 'Protein & Energy Requirements Fed for Every Critically ill patient every Time' (PERFECT) protocol: a before-and-after study.)
9.255. Malnutrition causes loss of lean body mass a determinant of outcome with the early protein deficit being an important contributor to the loss. The positive response of prealbumin to higher protein delivery shows that increasing proteins delivery is a treatment option.
This is correct. Two prospective studies might reinforce this observation and these references should be added at this point: 1) Critical illness is characterized by a high degree of stress and accelerated degradation of proteins that cause malnutrition, systemic inflammation and organ dysfunction, with a significant association between albumin, ferritin and transferrin (Nutr Hosp. 2015 Dec 1;32(6):2848-54. Imbalances in protein metabolism in critical care patient with systemic inflammatory response syndrome at admission in intensive care unit). 2) Only 22.7% of patients without protein deficiencies versus 37% of those at risk or already deficient developed multiple-organ system failure. Transferrin and prealbumin levels improved at the end of the period of early enteral feeding, while survivors had higher prealbumin levels than non-survivors (Nutrition. 2001 Jul-Aug;17(7-8):548-57. Malnutrition, nutritional indices, and early enteral feeding in critically ill children).
10.291. In randomized trials, our group showed that, in case of insufficient enteral feeding, supplemental PN guided by indirect calorimetry reduces nosocomial infections [12] and related costs [31].
Please at this point enrich your text and references with the following concluding remarks, based on studies using indirect calorimetry and including a high percentage of CCI patients: 1) A two-phase approach for nutritional support may more appropriately account for the physiologic changes during critical illness than one-phase approach (Curr Opin Crit Care. 2018 Aug;24(4):262-268. Trophic or full nutritional support?) 2) Nutrition is more than the sum of its parts (Pediatr Crit Care Med. 2018 Nov;19(11):1087-1089. Nutrition Is More Than the Sum of Its Parts), and most importantly 3) If you get good nutrition, you will become happy and if you get a bad one, you will become an ICU researcher (Pediatr Crit Care Med. 2019 Jan;20(1):89-90. If You Get Good Nutrition, You Will Become Happy; If You Get a Bad One, You Will Become an ICU Philosopher).
Author Response
We sincerely thank the 2 reviewers for high quality, and rapid review of our manuscript, and for the many great suggestions.
Reviewer #1
The study is wall organized and written, having suffered from limitations, which authors adequately report at the end of their discussion. I have some important suggestions or comments to the authors, which I believe, if answered, will improve their article and enhance their messages to the readers.
Thank you for these positive comments and for the below suggestions which have all been integrated
1.37. It is characterized by lengthy hospital stays, ….. [2].
Chronic critical illness (CCI) has also been characterized by an increasing trend of recognized genetic influence. Please add this info and the reference [1].
We agree with this suggestion and have inserted the proposed reference
1.38. Mechanical ventilation, …. [3].
It is important to refer to a study showing that chronic comorbid illnesses have been also identified as strong independent predictors of this condition [2]
We gain agree with this suggestion and have completed the text accordingly, and inserted the proposed reference
5.142. Table 2.
Energy presents as a balance (Cumulated Energy balance D10 (vs prescription)) on D10 while protein as intake only on D1-10 confusing the reader (Protein delivery D1-D10 (median [IQR]) g/day). It could be better to follow the same rules for energy and protein, preferably balance for both. In Table 2 we do not know if more protein has been prescribed than delivered
Indeed this was confusing. We have addressed this issue by completing Table 2, and by following the same rules (day partitions) for protein and energy. The new Table 2 shows energy and protein delivery for the first 10 days and from days 11 to 30. We also added the cumulated protein balance for the first 10 days and day 1 to 30. Contributing to the confusion we had been using the wording “prescription” instead of “delivery” in the figure – which has been corrected.
Overall | Non-survivors | Survivors | p | ||||
N (%) | 150 | 27 (18%) | 123 (82%) | ||||
Days of fasting : N per patient (median [IQR]) | 2.0 [1.0, 3.0] | 3.0 [1.0, 4.0] | 2.0 [1.0, 3.0] | 0.043 | |||
Percentage of days (median [IQR]) | 5.4 [2.4, 10.0] | 7.8 [4.3, 10.9] | 4.9 [2.0, 9.3] | 0.031 | |||
Prealbumin (delta of stay ) g/L | 0.07[0.04-0.12] | 0.06[0.02-0.10] | 0.07[0.04-0.13] | <0.001 | |||
Energy delivery D1-10 (median [IQR]) kcal/day (median [IQR]) Kcal/kg/day | 1161[957,1370] 15.8[11.8-18.9] | 1121[936,1385] 16.4[12.1-17.9] | 1161[983,1368] 15.7[11.8-19.1] | 0.719 0.838 | |||
Energy delivery D11-30 (median [IQR]) kcal/day (median [IQR]) Kcal/kg/day | 1559[1368-1762] 20.8[17.9-23.9] | 1504[1284-1645] 20.2[15.5-21.9] | 1581[1387-1772] 20.9[18.1-24.1] | 0.104 0.151 | |||
Cumulated Energy balance D1-10 (median [IQR]) kcal/day (median [IQR]) kcal/kg/day | -5266 [-8365, -2697] -70 [-102,-37] | -5365 [-9208, -2852] -74 [-125, -38] | -5234 [-8043, -2651] -69 [-101, -36 ] | 0.519 0.345 | |||
Cumulated Energy balance D1-30 (median [IQR]) kcal /d (median [IQR]) kcal /kg/day | -7700[-11607,-4702] -96.8[-148.2,-59.7] | -7710[-12197,-5097] -92.4[-151.1,-67.1] | -7677[-11350,-4554] -97.5[-146.9,-58.4] | 0.532 0.801 | |||
Protein delivery D1-10 (median [IQR]) g/day (median [IQR]) g/kg/day | 53.7 [40.7, 64.3] 0.69 [0.52,0.86] | 54.0 [40.2, 61.9] 0.73 [0.57, 0.86] | 53.4 [41.7, 65.7] 0.68 [0.50, 0.87] | 0.673 0.768 | |||
Protein delivery D11-30 (median [IQR]) g/day (median [IQR]) g/kg/day | 75.4 [62.0, 90.3] 1.0 [0.6, 1.4] | 70.0 [49.4, 83.7] 0.95 [0.68, 1.1] | 76.4 [63.3, 90.4] 1.0 [0.8, 1.2] | 0.051 0.104 | |||
Cumulate protein balance D1-10(median [IQR]) g/day (median [IQR]) g/kg/day | -374[-595,-223] -4.86[-6.82,-3.36] | -352[-538,-244] -4.63[-6.18,-3.49] | -379[-608,216] -4.98[-7.07,-3.34] | 0.803 0.938 | |||
Cumulate protein balance D1-30 g/day (median [IQR]) g/kg/day | -603[-1070,-304] -7.91[-12.47,-4.13] | -531[-1006,-355] -7.89[-13.62,4.74] | -611[-1069,-299] -8.43[-12.34,-4.11] | 0.912 0.816 | |||
5.144. Figure 2. Evolution of the route of feeding over time….
According to the legend fasting days are presented and differences between survivors and non-survivors. I can see neither of these two. Also, where is PN (parenteral nutrition)? IV usually refers to a simple intravenous solution (Gl5%, NaCl 0.9% with electrolytes, etc). Abbreviations should be defined at the end of the legends.
You are of course correct: this was confusing.
The legend, which was incomplete due to not translating the French abbreviations (IV was PN) this has now been corrected. The acronyms refer to the predominant feeding route, but not to dextrose, NaCl or propofol (now mentioned in the methods). We have completed the legend: these are the data of all patients presented as % of all patients over the first 30 days.
Legends have been added to the axes.
The figure does not address any difference between survivors and non-survivors. The legend of Table 2 was erroneously added due to copy/past maneuver at the wrong place – it has been repositioned
6.149. Figure 3. Individual prescription of energy, and administration of proteins and glucose by day
Where is energy? Also, many patients were receiving each day 0 protein and 0 glucose (g/day). Are you sure? Something is wrong with this figure: could these recordings have actually not been by day but at a moment “snapshots” (i.e. 8 a.m.) of a day? You should re-check your data and even delete this figure.
Sorry again. This was not prescription but delivery. Please see what the prescriptions were like – the prescribed daily target was quite ok – 1750 kcal as mean and median value – with adaptation to individual body weights. Total energy delivery was as variable as that of proteins and glucose: we decided to focus on those 2 macro-substrates, providing energy delivery in Figure 4 (please see next answer). And yes, the deliveries were really zero for proteins on some days (exceptionally for glucose): this information goes lost if clean box plots had been used – the erratic delivery is probably something that should avoided . This graph shocked the team members, making them finally aware about our more than suboptimal management. One of the internal aims of this quality study was thereby achieved. Figure 3 has been modified to respect the 10 day partition of the other data (previously 14 days)
|
7.166. Figure 4. Energy balance, energy and protein intake …..
Again, one is balance the other intake! One goal is prescribed the other is not prescribed but fixed value. Energy intake is not presented as the legend indicates.
Thank you for indicating this systematic misleading information: legend has been corrected and adapted to the new figure 4 providing protein and energy delivery, AND energy balance in the lower box.
8.188. Figure 5. Evolution of blood glucose,
Units should be shown
The presentation of the units has been improved: units were provided on the y axis. The legend has also been completed with the units
9.242. The non survivors were characterized by a higher percentage of fasting days, ….
It is of worth to refer here to recent studies showing that volume-based feeding safely and effectively improves energy and protein delivery to critically ill patients compared to traditional rate-based feeding and might improve patient outcomes [3].
Thank you for the suggestion which has been inserted
9.255. Malnutrition causes loss of lean body mass a determinant of outcome …..
This is correct. Two prospective studies might reinforce this observation and these references should be added at this point: 1) Critical illness is characterized by a high degree of stress and accelerated degradation of proteins that cause malnutrition, systemic inflammation and organ dysfunction, with a significant association between albumin, ferritin and transferrin [4]. 2) Only 22.7% of patients without protein deficiencies versus 37% of those at risk or already deficient developed multiple-organ system failure. Transferrin and prealbumin levels improved at the end of the period of early enteral feeding, while survivors had higher prealbumin levels than non-survivors [5].
Thank you for the suggestions: the discussion has been completed accordingly with the references. Thank you for signaling them J.
10.291. In randomized trials, our group showed that, … [12] and related costs [31].
Please at this point enrich your text and references with the following concluding remarks, based on studies using indirect calorimetry and including a high percentage of CCI patients: 1) A two-phase approach for nutritional support may more appropriately account for the physiologic changes during critical illness than one-phase approach (Curr Opin Crit Care. 2018 Aug;24(4):262-268. Trophic or full nutritional support?) 2) Nutrition is more than the sum of its parts [6], and most importantly 3) If you get good nutrition, you will become happy and if you get a bad one, you will become an ICU researcher [7]
Thank you for the suggestion. We agree that the concluding remarks were not strong enough: your suggestion was integrated. The two phase approach was actually applied in our Swiss SPN trial, but was not strongly enough presented and your suggested references have been inserted.
But among the suggested references, 4 are from the same group and concern pediatric patients, and not adults. We have focused on the publications with data, and left out the last suggestion, despite its attractive title.

Reviewer 2 Report
Summary of research: Overall, this study aims to improve the outcome of long-stay ICU patients by examining hospital feeding practices. The findings are interesting and could translate to improvements in patient treatment. However, this paper requires significant revision to address the issues outlined below. Further, this paper would benefit from a revision aiming for brevity, clarity, and improved grammar. I suggest the authors refrain from overusing ":", avoid informal terms, and consult with or hire a copyeditor.
Issues to address
The core finding of this paper relates to the NRS score as a predictor for patient risk. Please define and describe the NRS in the introduction section of this paper before you abbreviate it. Also, the term "poor outcome" needs to be defined more clearly.
It's not clear what the authors mean by "long-stay program". Are there specific parameters that define long-stay at this particular hospital?
Line 16-17: What does "daily intakes" refer to? Define explicitly what blood glucose values were measured. What do you mean by "ICU and hospital stay"? Does this refer to the number of days spent in either the ICU or hospital as a total? What does 90 days refer to?
Line 18: There is a closed bracket at the end of "the first 10 days" that should be removed. Describe what was tested. Information about parametric and non-parametric testing should be moved to the body of the paper under "Methods".
Line 19: Please indicate what parameters are being used to describe length of ICU stay? Square brackets are not needed and are confusing. For example you could write: "The median (minimum, maximum) length of stay was 31 (26, 46) days. Only one value is needed to report IQR. Add information about mortality to this sentence.
Line 20: Start a new sentence ... "Non-survivors were older (p=0.024), tended toward a higher SAPSII score (p=0.072), and had significantly higher NRS scores (p=0.033).
Line 22: below what recommendation?
Line 23: remove "of days". Describe the proportion of days with fasting in survivors vs. non-survivors that accompanies the statistical result (p=0.038) you provided. Provide a statistic for the association between protein intake and prealbumin.
Line 32-34: Add "unit" after "intensive care". Please provide a reference for this statement.
Line 36-39: "denomination" is not an appropriate word for this statement. How are lengthy hospital stays and high mortality rates defined in this case?
Line 42: It's not clear what "metabolic handicap" means. Please use precise language or omit this statement completely.
Line 44: Define SOFA before you abbreviate it.
Line 47: Define NUTRIREA-2 before you abbreviate it.
Line 48: replace "in septic shock" with "experiencing septic shock" or "with septic shock". The colon is not necessary here. Start a new sentence... "Further, many patients are at risk of refeeding syndrome, ..." explain what this is.
Line 48: Remove "addressed this issue". How is "good outcomes" defined?
Line 52: Remove the colon. I'm not clear on what "ramping up" means and how this guideline relates to the findings of the Brazilian study.
Line 53-56: provide a reference. Define ESPEN.
Line 62-64: Describe what is meant by long-stay program. What hospital did this study take place in?
Line 70-71: put "included" before the colon.
Line 74: Explicitly state what blood glucose values were measured.
Line 76: put abbreviation in brackets ... ex: discharge medical research council (MRC) force score.
Line 80: define NUTSIA.
Line 81-84: "as a first line" ... for what? Remove colon. This section could be re-written with better flow and clearer writing.
Line 85: Unclear what is meant by this sentence.
Line 89: What is considered a nutritional energy source and a non-nutritional source? Please clarify this in Section 2.3 Nutrition protocol.
Line 93: what was the protein critical cutoff value expressed in g/kg/d?
Line 99-101: Please indicate how grade 3, 2, and 1 are evaluated. How are these grades scored with a maximal value of 60? Please explain.
Line 103-110: How were the data evaluated for normality? Describe the non-parametric tests used for data that were not normally distributed. Please indicate how calculations for the Kaplan-Meier analysis were made. Why are you showing the 25th and 75th percentiles? Simplify by showing median (IQR). Please indicate which analyses were done with each statistical program. Was a sample size calculation made?
Tables 1 & 2: Why is the IQR represented by two numbers? IQR = Q3-Q1. What statistical test was used to calculate the differences between groups? Please include this information in the table footing or include the statistcal value in the table.
Figure 1: The correct term is "Kaplan–Meier" analysis (https://www.tandfonline.com/doi/abs/10.1080/01621459.1958.10501452) Y-axis should read "Probability of Survival (%)" as survival probabilty would be represented by values 0 and 1.
Figure 2: Please describe the legend abbreviations in the figure caption. Can the authors remove the grey horizontal lines in the background? Is this suppose to represent "EN"? What does the white upside down triangle at day 1 represent?
Figure 3: I haven't come across the term "spaghetto-gram" in any literature or other academic resources. Is this a legitimate term? What is the value in showcasing individual data rather than box and whisker plots, which are easier to interpret.
Line 198-199: Is there any indication why energy and protein intakes vary so greatly across patients? Is there a standard protocol for ICU patients established by the hospital?
Line 205-206: Why were there so many cases of ESPEN violations in your study?
Line 209-210: change to "Thereafter, the patients were better fed, but the majority did not reach their prescribed goals".
Line 211-212: Was this significantly higher?
Line 213: How does 35.3% compare to the general ICU mortality?
Line 218-219: As-written, this sentence is difficult to read. Please re-state for clarity.
Line 225-227: This statement can be shortened. Remove "why was that?"
Line 230-231: This is re-stating what written in the introduction almost verbatim. Re-stating is redundant.
Line 234: what is "52'563"? Typo?
Line 234-236: It's not necessary to include so much detail of each trial
Line 241-243: Was there a difference in the types of illness found in non-survivors vs. survivors. What accounted for a higher proportion of fasting days? You provide a general reference (ref#20) but it would be of interest to the reader to know more details about the current population studied. Non-survivors were significantly older but you did not talk about this.
Line 247: What is the point of mentioning the title of the study? Please remove this.
Line 249: remove the apostrophe from "-6'000 kcal"
Line 252-254: It's not clear whether the authors have verified these statements using statistical methods. Why wasn't this reported in Table 2?
Line 265: "based the results this too late" ... grammatical error.
Line 273: Is "long-stayers" a technical term?
APPENDIX
Figure A/B: Can you indicate which time points were significant? If there were no significant findings, report this in the figure caption.
Author Response
REVIEWER #2
We thank the reviewer for a careful revision, and multiple, constructive and pragmatic suggestions and questions.
The core finding of this paper relates to the NRS score as a predictor for patient risk. Please define and describe the NRS in the introduction section of this paper before you abbreviate it. Also, the term "poor outcome" needs to be defined more clearly.
It's not clear what the authors mean by "long-stay program". Are there specific parameters that define long-stay at this particular hospital?
We have addressed this definition issue with more details and provided the definition to “of chronically critically ill (CCI) patients defined by an ICU stay > 2 weeks”. We hope that using a well know appellation addresses your comment.
We have also more clearly defined what this program called “PLS” (patients long séjour) included
Line 16-17: What does "daily intakes" refer to? Define explicitly what blood glucose values were measured. What do you mean by "ICU and hospital stay"? Does this refer to the number of days spent in either the ICU or hospital as a total? What does 90 days refer to?
To address your comment we have completed the definitions:
"daily intakes" has been expanded to “total daily energy from nutritional and non-nutritional sources , protein and glucose intakes”
Blood glucose – were determined on the blood gas analyzer on arterial samples
"ICU and hospital stay":” length of” has been added to complete
“90 days” refers to time of follow up (after admission) and refers to survival
Line 18: There is a closed bracket at the end of "the first 10 days" that should be removed. Describe what was tested. Information about parametric and non-parametric testing should be moved to the body of the paper under "Methods".
Thank you: Has been removed
Line 19: Please indicate what parameters are being used to describe length of ICU stay? Square brackets are not needed and are confusing. For example you could write: "The median (minimum, maximum) length of stay was 31 (26, 46) days. Only one value is needed to report IQR. Add information about mortality to this sentence.
Modified, but we have kept the IQR 25-75 (Q1,Q3) values due to the very unsymmetrical distribution. Please also see our answer to your comment Line 103-110
Line 20: Start a new sentence ... "Non-survivors were older (p=0.024), tended toward a higher SAPSII score (p=0.072), and had significantly higher NRS scores (p=0.033).
Thank you: text was modified as suggested
Line 22: below what recommendation?
This relates to the ICU’s feeding protocol – the sentence has been reworded
Line 23: remove "of days". Describe the proportion of days with fasting in survivors vs. non-survivors that accompanies the statistical result (p=0.038) you provided. Provide a statistic for the association between protein intake and prealbumin.
Thank you. The number of days has been provided in Table 2.
Prealbumin data have been included: the correlation between protein intakes and increase in prealbumin was R2 =0.19 p = 0.027. We have included in Table 2, the delta-prealbumin value (difference between first and last value of the ICU stay), which differed significantly between survivors and non survivors.
Line 32-34: Add "unit" after "intensive care". Please provide a reference for this statement.
“Unit” was inserted. We used Nelson et al [8] to support the first sentence
Line 36-39: "denomination" is not an appropriate word for this statement. How are lengthy hospital stays and high mortality rates defined in this case?
“Denomination” has been replaced “designation”. CCI refers to the ICU stay and may progress to prolonged hospital stay
Line 42: It's not clear what "metabolic handicap" means. Please use precise language or omit this statement completely.
We have tried to reformulate – and searched the Merriam-Webster to find synonyms. We respectfully would like to maintain the wording “metabolic handicap”, but have clarified the sentence
Handicap is a precise, well accepted word to define a condition that creates difficulty for achieving success and makes progress difficult (please see below).
A metabolic handicap is a condition that has built up before the actual ICU admission (such as weight loss, comorbidities, age, or malnutrition) that hinders a full and optimal immune and metabolic response to the acute condition. It compromises survival.
Synonyms and Antonyms of handicap- according to Merriam-Webster
A feature of someone or something that creates difficulty for achieving success
her natural shyness was not a handicap when she played chess
Synonyms of handicap : debit, disadvantage, disbenefit, downside, drawback, incommodity, liability, minus, negative, strike
2 - Something that makes movement or progress difficult
her uncomfortable shoes became a handicap on the walking tour of the city, as she often had to sit and rest her sore feet
Synonyms of handicap : balk, bar, block, chain, clog, cramp, crimp, deterrent, drag, embarrassment, encumbrance, fetter, hindrance, holdback, hurdle, impediment, inhibition, interference, let, manacle, obstacle, obstruction, shackles, stop, stumbling block, trammel
Line 44: Define SOFA before you abbreviate it.
The definition was added as suggested.
Line 47: Define NUTRIREA-2 before you abbreviate it.
We understand the request but the title’s acronym is in fact the abbreviation of a French title “Nutrition artificielle en Réanimation 2”, while the official complete title of the publication mentions “randomised, controlled, multicentre, open-label, parallel-group study” – NUTRIREA is not the acronym of the English title - so we just feel it does not provide an information - was reformulated.
Line 48: replace "in septic shock" with "experiencing septic shock" or "with septic shock". The colon is not necessary here. Start a new sentence... "Further, many patients are at risk of refeeding syndrome, ..." explain what this is.
Thank you, we corrected the text as suggested in 2 places (introduction and discussion).
Line 48: Remove "addressed this issue". How is "good outcomes" defined?
Addressed has been deleted.
Couto et al defined the short term outcomes (duration of mechanical ventilation, length of ICU stay and mortality) and long-term (functional capacity and mortality) clinical outcomes. Has been inserted
Line 52: Remove the colon. I'm not clear on what "ramping up" means and how this guideline relates to the findings of the Brazilian study.
The Merriam-Webster definition of “ramp up” is: “an increase in the amount of products …”
Ramping up has become a wording used by the ESPEN guideline group, which means increasing progressively the nutrients over 3-4 days depending on the patient’s condition. The aim is to prevent full feeding for the explained reasons. Absence of progression with full early feeding (Couto: more than 70% of the target nutrition in 72 hours), is possible a reasons why Couto et al observed disappointing results. The sentence has been expanded, and a reference inserted to support the concept [9].
Line 53-56: provide a reference. Define ESPEN.
The acronym was defined: European Society of Clinical Nutrition and Metabolism (ESPEN) – the acronym was based on the first name that was modified about 15 years ago.
Line 62-64: Describe what is meant by long-stay program. What hospital did this study take place in?
Description has been completed. The study took place in our Lausanne university hospital’s multidisciplinary ICU
Line 70-71: put "included" before the colon.
Done.
Line 74: Explicitly state what blood glucose values were measured.
Only arterial blood glucose determined on a point-of-care apparatus was used: inserted
Line 76: put abbreviation in brackets ... ex: discharge medical research council (MRC) force score.
Thank you: done.
Line 80: define NUTSIA.
Done - a French acronym for “NUTrition aux Soins Intensifs Adultes”
Line 81-84: "as a first line" ... for what? Remove colon. This section could be re-written with better flow and clearer writing.
The colon was removed. We have clarified the sentence which now reads: “Enteral nutrition is recommended as the first option when nutritional therapy is indicated (patients in whom oral intake is not possible).”
Line 85: Unclear what is meant by this sentence.
We agree that this “Energy intakes include … (drug dilution and sedation)” was not clear, and we have reworded.
Line 89: What is considered a nutritional energy source and a non-nutritional source? Please clarify this in Section 2.3 Nutrition protocol.
Has been clarified: “nutritional energy source” refers to intakes that result from prescribed feeding, while non nutritional are those substrates coming with drug dilution fluids and fat coming from the sedative propofol. Our computerized information system has been customized to integrate all. This is why we have also customized a specific item indicating the route of feeding - hence a patient receiving glucose due to drugs and fat from propofol is not considered “fed” if there is not prescribed nutrition.
Line 93: what was the protein critical cutoff value expressed in g/kg/d?
No, while the goal was 1.2 g/kg/day which was the value used to calculate the deficit, the cut off was the sum of the differences between the ideal goal and the quantity delivered, based on the study by Yeh et al [10].
Line 99-101: Please indicate how grade 3, 2, and 1 are evaluated. How are these grades scored with a maximal value of 60? Please explain.
Thank you for this important observation. We have detailed the grading system of MRC score.
Six muscle groups were bilaterally measured (abduction of the arm, flexion of the forearm, extension of the wrist, flexion of the hip, extension of the knee, and dorsal flexion of the foot). All muscle groups were scored between 0 and 5 (0 no visible/palpable contraction; 1 visible/palpable contraction without movement of the limb; 2 movement of the limb but not against gravity; 3 movement against gravity (almost full passive range of motion) but not against resistance; 4 movement against gravity and resistance, arbitrarily judged to be submaximal for gender and age; 5 normal). The maximal score is hence 60 points.
Line 103-110: How were the data evaluated for normality? Describe the non-parametric tests used for data that were not normally distributed. Please indicate how calculations for the Kaplan-Meier analysis were made. Why are you showing the 25th and 75th percentiles? Simplify by showing median (IQR). Please indicate which analyses were done with each statistical program. Was a sample size calculation made?
The majority of variables had a Poisson distribution, i.e. very asymmetrical.
Normal distribution was evaluated by visual assessment of distribution histograms and calculation of skewness and kurtosis. For non-parametric variables, the Kruskal-Wallis test was used.
For the Kaplan-Meier analysis, the log rank test was used to compare full curves of each group.
We agree that the reporting you suggest is simpler to read, but as the distribution of the values was really not normal, we prefer to maintain the Q1 and Q3 reporting. Indeed one limitation of reporting the IQR as one value is that the IQR might be either symmetrical or asymmetrical around the median. The statistical paragraph has been completed.
Tables 1 & 2: Why is the IQR represented by two numbers? IQR = Q3-Q1. What statistical test was used to calculate the differences between groups? Please include this information in the table footing or include the statistcal value in the table.
For parametric variables we used the one-way test and for non-parametric the Kruskal-Wallis Rank Sum Test. Information was included in the statistical section.
Figure 1: The correct term is "Kaplan–Meier" analysis (https://www.tandfonline.com/doi/abs/10.1080/01621459.1958.10501452) Y-axis should read "Probability of Survival (%)" as survival probabilty would be represented by values 0 and 1.
Thank you – sorry for the terrible typo: corrected, as well as the Y-axis legend
Figure 2: Please describe the legend abbreviations in the figure caption. Can the authors remove the grey horizontal lines in the background? Is this suppose to represent "EN"? What does the white upside down triangle at day 1 represent?
The abbreviations have been inserted in the capitation. The grey thin lines visible in the EN field were left to facilitate reading the scale.
The white triangle was generated by the inclusion of a day zero in the XLS base, which has been deleted – it had no significance on its own. Legends have been added to the x and y axes.
Figure 3: I haven't come across the term "spaghetto-gram" in any literature or other academic resources. Is this a legitimate term? What is the value in showcasing individual data rather than box and whisker plots, which are easier to interpret.
We respectfully request to maintain this presentation. The “spaghetto-gram” appellation is a neologism of my own, and is therefore in brackets: the analogy is with a bowl of cooked spaghetti that spread in all directions if they escape.
The aims of this figure was to show a phenomenon which is not captured by box-plots and whiskers, namely the extreme day to day variability that characterized the nutrition of all these patients. The patients are never the mean value: not integrating that one important problem is this variability: talking about the mean or median values confirms a sense of normality that does not correspond to what the patients’ experience. Look at the means and median values that are provided: the values look reasonable, although low, and rather normal, which it was not the case at the individual level. It reflects the multiple interruptions due to multiple sorts of procedures that these patients needed.
Your co-reviewer reacted to the occurrence of the zero values affecting proteins more than glucose, showing the importance of this figure.
Line 198-199: Is there any indication why energy and protein intakes vary so greatly across patients? Is there a standard protocol for ICU patients established by the hospital?
There is indeed a protocol in the ICU (called NUTSIA as mentioned in the text), which has been validated by the staff physicians, nurses and dieticians. But we humbly have to admit that several physicians do not apply it, for variable good and bad reasons. This quality study has one important merit: to show the team that we have a problem and must address it – the improvement process has started.
Line 205-206: Why were there so many cases of ESPEN violations in your study?
This is indeed a paradox as MMB is a member of the ESPEN-ICU guidelines group. It reflects globally different problems: 1) the complexity of these patients who were difficult to feed enterally, 2) the non-compliance with the protocol by some specialists, 3) the refusal to apply of the concept of combined EN+PN feeding or even of PN because still scared by what was published with EPaNIC and a few other trials, 4) the insufficient presence of dieticians
Line 209-210: change to "Thereafter, the patients were better fed, but the majority did not reach their prescribed goals".
Was changed as suggested
Line 211-212: Was this significantly higher?
Yes the score and mortality were significantly higher
Line 213: How does 35.3% compare to the general ICU mortality?
It was significantly higher - has been written
Line 218-219: As-written, this sentence is difficult to read. Please re-state for clarity.
We hope reformulation reads better
Line 225-227: This statement can be shortened. Remove "why was that?"
Was shortened and “..” deleted
Line 230-231: This is re-stating what written in the introduction almost verbatim. Re-stating is redundant.
We agree that redundancy should be avoided, but this is an important point that we really want to emphasize in the discussion: I have been surprised during congresses at how many clinicians are not aware. We have reworded and shortened to address your comment.
Line 234: what is "52'563"? Typo?
Thank you for seeing the typo it was “52,563”, i.e. fifty two thousand five hundred and sixty three patients.
Line 241-243: Was there a difference in the types of illness found in non-survivors vs. survivors. What accounted for a higher proportion of fasting days? You provide a general reference (ref#20) but it would be of interest to the reader to know more details about the current population studied. Non-survivors were significantly older but you did not talk about this.
The paragraph has been shortened.
Yes, the non-survivors were indeed older. As in other studies [8], age is a component of frailty, and risk factor for becoming CCI: our data just confirm it. This has been addressed (lines 334-335).
Line 247: What is the point of mentioning the title of the study? Please remove this.
The point is that the authors exactly showed that! We have modified the sentence to contain the meaning - and avoid the criticism of plagiarism
Line 249: remove the apostrophe from "-6'000 kcal"
Thank you. It was corrected to -6,000 kcal.
Line 252-254: It's not clear whether the authors have verified these statements using statistical methods. Why wasn't this reported in Table 2?
Yes this statement had been verified – is now reported in Table 2
Line 265: "based the results this too late" ... grammatical error.
Sorry this was an incomplete sentence, which has been rewritten
Line 273: Is "long-stayers" a technical term?
This was the translation of our French designation of these patients (patients long séjour). CCI is the most frequently used appellation in the English literature. Has been modified.
APPENDIX Figure B/C (former A/B): Can you indicate which time points were significant? If there were no significant findings, report this in the figure caption.
We have indicated in the capitation what was significant. We did a GLM analysis (Generalized linear model: this information has been included in the statistical methods). But indicating one single point difference with repeated analysis require an equivalent of Bonferroni correction (p value * 30 in our case), so we did not test individually the different time points, but provide the results of the two-way ANOVA.
A figure A has been inserted in the Web-Appendix which shows the evolution of the SOFA scores and shows that these patients were really sick through their stay.
References
1. Briassoulis G; Filippou O; Mavrikiou M; Natsi L; Ktistaki M; Hatzis T. Current trends of clinical and genetic characteristics influencing the resource use and the nurse-patient balance in an intensive care setting. Journal of critical care 2005, 20, 139-146, doi:10.1016/j.jcrc.2005.04.003.
2. Frengley JD; Sansone GR; Kaner RJ. Chronic Comorbid Illnesses Predict the Clinical Course of 866 Patients Requiring Prolonged Mechanical Ventilation in a Long-Term, Acute-Care Hospital. J Intensive Care Med 2018, e-pub, 885066618783175, doi:10.1177/0885066618783175.
3. Brierley-Hobson S; Clarke G; O'Keeffe V. Safety and efficacy of volume-based feeding in critically ill, mechanically ventilated adults using the 'Protein & Energy Requirements Fed for Every Critically ill patient every Time' (PERFECT) protocol: a before-and-after study. Critical care 2019, 23, 105, doi:10.1186/s13054-019-2388-7.
4. Bouharras El Idrissi H; Molina Lopez J; Perez Moreno I; Florea DI; Lobo Tamer G; Herrera-Quintana L; Perez De La Cruz A; Rodriguez Elvira M; Planells Del Pozo EM. Imbalances in Protein Metabolism in Critical Care Patient with Systemic Inflammatory Response Syndrome at Admission in Intensive Care Unit. Nutricion hospitalaria 2015, 32, 2848-5284, doi:10.3305/nh.2015.32.6.9827.
5. Briassoulis G; Zavras N; Hatzis T. Malnutrition, nutritional indices, and early enteral feeding in critically ill children. Nutrition 2001, 17, 548-557.
6. Briassoulis G; Briassoulis P; Ilia I. Nutrition Is More Than the Sum of Its Parts. Pediatr Crit Care Med 2018, 19, 1087-1089.
7. Briassoulis G; Briassoulis P; Ilia S. If you get Good Nutrition, you will become happy; If you get a bad one, you will become an ICU philosopher. Pediatr Crit Care Med 2019, 20, 89-90, doi:10.1097/PCC.0000000000001774.
8. Nelson JE; Cox CE; Hope AA; Carson SS. Chronic critical illness. American journal of respiratory and critical care medicine 2010, 182, 446-454, doi:10.1164/rccm.201002-0210CI.
9. Oshima T; Berger MM; De Waele E; Guttormsen AB; Heidegger CP; Hiesmayr M; Singer P; Wernerman J; Pichard C. Indirect calorimetry in nutritional therapy. A position paper by the ICALIC study group. Clinical nutrition 2017, 36, 651-662, doi:10.1016/j.clnu.2016.06.010.
10. Yeh DD; Fuentes E; Quraishi SA; Cropano C; Kaafarani H; Lee J; King DR; DeMoya M; Fagenholz P; Butler K, et al. Adequate Nutrition May Get You Home: Effect of Caloric/Protein Deficits on the Discharge Destination of Critically Ill Surgical Patients. JPEN. Journal of parenteral and enteral nutrition 2016, 40, 37-44.
Round 2
Reviewer 1 Report
Much improved. Some text editing is still needed one verification, and two corrections of inconsistencies presented below.
Minor concerns.
Figure 3. Individual prescription of energy, and administration of proteins and glucose by day… And yes, the deliveries were really zero for proteins on some days (exceptionally for glucose): this information goes lost.
• Again. Are you sure there were so many patients receiving zero protein so many days and even patients receiving zero glucose? Could this be an error entered either during the figure production or of data entered in the database? Sometimes missing values are replaced by zero in statistical programs. In their response authors present another figure (energy?) showing zero delivery on day 1 only. If this is energy provided by EN or PN, in these formulae a major constituent is carbohydrates which are provided daily. The same with protein. How could this be possible? I think some clarification is needed here.
11.326. “Supporting this hypothesis, Bouhadras 326 et al [29] showed that only 22.7% of patients without protein deficiencies, versus 37% of those at risk or 327 already deficient, developed multiple-organ system failure. Further transferrin and prealbumin levels 328 improved at the end of the period of early EN, and survivors had higher prealbumin levels than non-329 survivors [30].”
• The info provided is not correct. Although authors state that: the discussion has been completed accordingly with the references, typo misplacements have been introduced. Obviously, reference 30 which has shown these findings (22.7%... 37%) has been misplaced to the end of the sentence as ref 30, while a previous, non-relevant reference 29, has been erroneously re-entered here. Thus, the real reference 31 at the end of the sentence has been omitted! Please substitute the correct references as shown below, and renumber references downwards.
• Supporting this hypothesis, Briassoulis 326 et al [29] showed that only 22.7% of patients without protein deficiencies, versus 37% of those at risk or 327 already deficient, developed multiple-organ system failure [30]. Further transferrin and prealbumin levels 328 improved at the end of the period of early EN, and survivors had higher prealbumin levels than non-329 survivors [30]. [31 = J Nutr Biochem. 2002 Sep;13(9):560. Influence of an aggressive early enteral nutrition protocol on nitrogen balance in critically ill children]. = this is the correct reference (now 31), which should be added here. Reorder references downwards.
11.339. The timing, dose and route of feeding clearly must be addressed more stringently and earlier, 339 with our dieticians assisting the clinical team in the high risk patients already by day3-4.
• The role of dieticians appears here for the first time, assuming their contribution is not quite productive! But this should not be appearing for the first time in the discussion without being mentioned in the methods and quantified in the results (a row in a table for example). I suggest to delete this sentence or to insert a relative reference of your own publications, in which such a conclusion has been reached.
Author Response
REVIEWER #1
Much improved. Some text editing is still needed one verification, and two corrections of inconsistencies presented below.
We are grateful for this positive comment and careful re-review: we have addressed below all the issues.
Minor concerns.
Figure 3. Individual prescription of energy, and administration of proteins and glucose by day… … the deliveries were really zero for proteins on some days (exceptionally for glucose): this information goes lost.
• Again. Are you sure there were so many patients receiving zero protein so many days and even patients receiving zero glucose? Could this be an error entered either during the figure production or of data entered in the database? Sometimes missing values are replaced by zero in statistical programs. In their response authors present another figure (energy?) showing zero delivery on day 1 only. If this is energy provided by EN or PN, in these formulae a major constituent is carbohydrates which are provided daily. The same with protein. How could this be possible? I think some clarification is needed here.
Sorry to have to disappoint you with the confirmation that these data have been verified several times manually, including the most threatening “zero glucose” (luckily the patients have some degree of endogenous glucose production). Zeroes are not a replacement for missing values. Our CIS (MetaVision) captures every mg of substrate from any dilution solution, and adds it to any substrate from feeding. Please see the below screen shot which shows the 24hr intakes on 6 consecutive days: the lines Protein24H and Glucides24H show total of 0g, and 1g respectively for the first 3 days (patient was on NaCl 0.9% and propofol-lipid). EN was first introduced after 3 days which is reflected by the appearance of positive values thereafter. These zero values were one of the important findings of this work – the problem has been addressed, and should not occur anymore.
The legend of the figure has been re-written (shortened).
11.326. “Supporting this hypothesis, Bouharras et al [29] showed that only 22.7% …., developed multiple-organ system failure. Further transferrin and prealbumin …. [30].”
• The info provided is not correct. Although authors state that: the discussion has been completed accordingly with the references, typo misplacements have been introduced. Obviously, reference 30 which has shown these findings (22.7%... 37%) has been misplaced to the end of the sentence as ref 30, while a previous, non-relevant reference 29, has been erroneously re-entered here. Thus, the real reference 31 at the end of the sentence has been omitted! Please substitute the correct references as shown below, and renumber references downwards.
Thank you for seeing these misplacements, which have been corrected
• Supporting this hypothesis, Briassoulis et al [29] showed that only 22.7% of patients without protein deficiencies, versus 37% of those at risk or already deficient, developed multiple-organ system failure [30]. Further transferrin and prealbumin levels improved at the end of the period of early EN, and survivors had higher prealbumin levels than non-survivors [30]. [31 = J Nutr Biochem. 2002 Sep;13(9):560. Influence of an aggressive early enteral nutrition protocol on nitrogen balance in critically ill children]. = this is the correct reference (now 31), which should be added here. Reorder references downwards.
Same as previous comment, the references have been replaced, and one reference added. Due to the inclusion of the dietician paper in the methods [17], the numbers are not the initial ones.
11.339. The timing, dose and route of feeding clearly must be addressed more stringently and earlier, with our dieticians assisting the clinical team in the high risk patients already by day3-4.
• The role of dieticians appears here for the first time, assuming their contribution is not quite productive! But this should not be appearing for the first time in the discussion without being mentioned in the methods and quantified in the results (a row in a table for example). I suggest to delete this sentence or to insert a relative reference of your own publications, in which such a conclusion has been reached.
We agree, the mention should have come earlier. Indeed the dieticians are involved much earlier in care, but the PLS program only discusses the patients after 2 weeks (too late), which stresses the dietician role in the early phase. Role of dieticians has been added in methods as suggested (lines 139-141), and one references has been inserted [1].
The sentence has been reformulated to better express what was meant: the early advices of the dieticians should be applied when formulated, which was frequently not done: waiting until LBM is lost is too late (lines 322-326). This quality issue has been addressed

Reviewer 2 Report
I appreciate the care taken by the authors in making extensive corrections to this manuscript. I think it has been greatly improved and may represent an important contribution to the medical literature. Nevertheless, there are two persisting issues that I am unsatisfied with and feel must be corrected before publication.
You are still not expressing IQR correctly. Writing [IQR 25/75] in the text is not standard in the scientific literature. If you prefer to express Q1 and Q3, then state that. Otherwise, make the appropriate calculation for IQR (Q3-Q1) and insert these values into text. The current form is unacceptable. Further, I disagree with your premise that Q1 and Q3 are needed because of the mostly non-normal distribution of variables. IQR is appropriate for this type of data. This calculation enables the reader to assess the variability in the data better than having Q1 and Q3. Otherwise, the reader will have to do mental math in order to assess variability.
Regarding the use of "spaghetto-gram", I maintain that this informal personal term is not appropriate for scientific literature. There is no point in having this analogy. The reader can see that the data is erratic by looking at it. Please remove this term.
Author Response
REVIEWER #2
I appreciate the care taken by the authors in making extensive corrections to this manuscript. I think it has been greatly improved and may represent an important contribution to the medical literature. Nevertheless, there are two persisting issues that I am unsatisfied with and feel must be corrected before publication.
Thank you for acknowledging our efforts, and please see below our answers to your comments
You are still not expressing IQR correctly. Writing [IQR 25/75] in the text is not standard in the scientific literature. If you prefer to express Q1 and Q3, then state that. Otherwise, make the appropriate calculation for IQR (Q3-Q1) and insert these values into text. The current form is unacceptable. Further, I disagree with your premise that Q1 and Q3 are needed because of the mostly non-normal distribution of variables. IQR is appropriate for this type of data. This calculation enables the reader to assess the variability in the data better than having Q1 and Q3. Otherwise, the reader will have to do mental math in order to assess variability.
Sorry, but the expression of IQR was according to recommendations by our internal statisticians: there are different valid options to present interquartile ranges. But as Reviewer, you may impose your view.
As you left us a choice, we preferred the option Q1 and Q3, because the IQR is asymmetrical around the median, which would go unnoticed if reported as one value rather than a range. We made changes accordingly in the methods and results. The results were expressed as median (Q1, Q3) which is specified in the methods.
Regarding the use of "spaghetto-gram", I maintain that this informal personal term is not appropriate for scientific literature. There is no point in having this analogy. The reader can see that the data is erratic by looking at it. Please remove this term.
The term has been deleted, and the legend reformulated